# Calcitriol, an Active Form of Vitamin D3, Mitigates Skin Barrier Dysfunction in Atopic Dermatitis NC/Nga Mice

**DOI:** 10.3390/ijms24119347

**Published:** 2023-05-27

**Authors:** Yoshie Umehara, Juan Valentin Trujillo-Paez, Hainan Yue, Ge Peng, Hai Le Thanh Nguyen, Ko Okumura, Hideoki Ogawa, François Niyonsaba

**Affiliations:** 1Atopy (Allergy) Research Center, Juntendo University Graduate School of Medicine, Tokyo 113-8421, Japan; y-umeha@juntendo.ac.jp (Y.U.); t-valentin@juntendo.ac.jp (J.V.T.-P.); h-yue@juntendo.ac.jp (H.Y.); g-peng@juntendo.ac.jp (G.P.); ha-nguyen@juntendo.ac.jp (H.L.T.N.); kokumura@juntendo.ac.jp (K.O.); ogawa@juntendo.ac.jp (H.O.); 2Department of Dermatology and Allergology, Juntendo University Graduate School of Medicine, Tokyo 113-8421, Japan; 3Faculty of International Liberal Arts Global Health Studies, Juntendo University, Tokyo 113-8421, Japan

**Keywords:** calcitriol, atopic dermatitis, skin barrier function, tight junction, *Dermatophagoides farinae* body ointment-induced NC/Nga mouse model of atopic dermatitis, transepidermal water loss, tight junction permeability assay

## Abstract

Atopic dermatitis and psoriasis are prevalent chronic inflammatory skin diseases that are characterized by dysfunctional skin barriers and substantially impact patients’ quality of life. Vitamin D3 regulates immune responses and keratinocyte differentiation and improves psoriasis symptoms; however, its effects on atopic dermatitis remain unclear. Here, we investigated the effects of calcitriol, an active form of vitamin D3, on an NC/Nga mouse model of atopic dermatitis. We observed that the topical application of calcitriol decreased the dermatitis scores and epidermal thickness of NC/Nga mice with atopic dermatitis compared to untreated mice. In addition, both stratum corneum barrier function as assessed by the measurement of transepidermal water loss and tight junction barrier function as evaluated by biotin tracer permeability assay were improved following calcitriol treatment. Moreover, calcitriol treatment reversed the decrease in the expression of skin barrier-related proteins and decreased the expression of inflammatory cytokines such as interleukin (IL)-13 and IL-33 in mice with atopic dermatitis. These findings suggest that the topical application of calcitriol might improve the symptoms of atopic dermatitis by repairing the dysfunctional epidermal and tight junction barriers. Our results suggest that calcitriol might be a viable therapeutic agent for the treatment of atopic dermatitis in addition to psoriasis.

## 1. Introduction

Atopic dermatitis is one of the most common allergy-mediated inflammatory skin diseases and is characterized by relapsing eczema, skin dryness, and intense pruritus. Severe pruritus can disturb work and sleep in patients with atopic dermatitis, severely affecting their quality of life [1,2]. The pathogenesis of atopic dermatitis involves interactions between skin barrier impairment and abnormal immune responses featuring enhanced type 2 inflammation [3].

Dysfunction of the skin barrier increases transepidermal water loss (TEWL) and enhances the penetration of antigens, irritants, and pathogens. Skin barrier function is mostly maintained by two physical barriers, the stratum corneum and tight junctions, which are disrupted in patients with atopic dermatitis [3,4]. The stratum corneum is in the outermost part of the epidermis and is a barrier unique to the skin. During differentiation from epidermal keratinocytes to corneocytes, the cells become compact, and a cornified envelope that lines the cell membrane is formed from keratin, filaggrin, involucrin, and loricrin [5]. Filaggrin, a key structural protein in keratinocytes and the major component of keratohyalin granules is expressed in the stratum granulosum layer. Loss-of-function mutations within the filaggrin gene strongly predispose patients to atopic dermatitis, while the expression of filaggrin is reduced in patients with atopic dermatitis without filaggrin gene mutations [6]. Loricrin and involucrin are key structural proteins of the cornified envelope and are important for flexibility and mechanical strength in corneocytes [5,7]. These molecules coordinately control epidermal hydration, pH regulation, and microbial ecology, which are components of the skin barrier [3]. Tight junctions are present in other epithelia as well, and their main function is to connect epithelial cells and limit the penetration of allergens, microbes, and irritants, as well as the movement of water, ions, and solutions. The molecular makeup of tight junctions includes the claudin family, as well as occludin, junctional adhesion molecules, and *zonula occludens* (ZO) proteins [8]. Claudin-1 is the most characterized protein in tight junctions and is critically important for skin barrier function. *Claudin-1* knockout mice cannot survive more than 1 day after birth due to lethal dehydration [9,10,11]. Occludin is a transmembrane protein that was the first identified component of tight junctions [12]. ZO-1 is a cytoplasmic plaque protein that also composes tight junctions and connects to the intracellular domain of claudin-1 and occludin. Decreased levels of tight junction molecules are associated with the dysfunction of the skin barrier in atopic dermatitis [3,4].

To manage psoriasis, patients are treated with various drugs, including corticosteroids, vitamin D3 analogs, retinoids, methotrexate, and cyclosporin [13]. Among these drugs, corticosteroids, methotrexate, and cyclosporin are also frequently used for the treatment of atopic dermatitis [14]. Additionally, phototherapies such as ultraviolet B, psoralen, ultraviolet A (PUVA), and narrowband ultraviolet B (NB-UVB) are efficacious in the treatment of both psoriasis and atopic dermatitis [14]. The topical application of vitamin D3 analogs improves psoriasis symptoms by regulating immune responses and keratinocyte differentiation [15].

Vitamin D3 is a secosteroid that is crucial for the regulation of tissue homeostasis [16]. The association between vitamin D deficiency and allergy development has been demonstrated in many studies, and low vitamin D levels have been reported in patients with atopic dermatitis [16,17,18,19]. Several studies reported that oral supplementation with vitamin D significantly decreased severity scores in patients with atopic dermatitis [20,21]. In addition, the development of allergen-triggered eczema in a mouse model was inhibited by an intraperitoneal injection of a vitamin D receptor agonist [22]. Moreover, a case report has shown that topically applied calcitriol ameliorates atopic dermatitis of the eyelid [23].

These studies suggest the therapeutic benefits of vitamin D in atopic dermatitis in addition to psoriasis. However, the effects of vitamin D3 on atopic dermatitis remain unclear. The present study investigated the topical effects of calcitriol, an active form of vitamin D3, on a *Dermatophagoides farinae* body (Dfb) ointment-induced NC/Nga mouse model of atopic dermatitis.

## 2. Results

### 2.1. Topical Application of Calcitriol Improves the Skin Condition and TEWL in NC/Nga Mice with Atopic Dermatitis

We examined the effect of the topical application of calcitriol on dermatitis and TEWL in an NC/Nga mouse model of atopic dermatitis. NC/Nga mice spontaneously develop atopic dermatitis. These mice do not show skin lesions in specific-pathogen-free (SPF) conditions; however, when they are kept in non-air-controlled conventional conditions, they develop various grades of dermatitis from the age of 8 weeks [24]. NC/Nga mice develop atopic dermatitis under SPF conditions following sensitization to house dust mite allergens or haptens, such as 2,4-dinitrochlorobenzene (DNCB) and 2,4-dinitrofluorobenzene (DNFB), and picryl chloride [25]. To examine the effect of calcitriol on atopic dermatitis in NC/Nga mice, dermatitis was induced on the rostral back skin of NC/Nga mice using Dfb ointment, as shown in Figure 1. In our experimental conditions, a daily topical application of 1 nmol calcitriol to NC/Nga mice without dermatitis for 4 days did not affect the skin condition or TEWL in these mice compared with vehicle-treated mice (Figure 2a,d). Calcitriol gradually alleviated dermatitis and significantly decreased TEWL on Day 3 and Day 4 of treatment in NC/Nga mice with atopic dermatitis (Figure 2b,e). In addition, the effect of calcitriol on C57BL/6J mice was examined because the long-term application of calcipotriol, a vitamin D3 analog also known as MC903, has been frequently used for the induction of dermatitis in C57BL/6 mice [26,27,28,29,30]. Although the application of 1 nmol calcitriol on C57BL/6J mice caused slight redness from Day 2 until Day 4, neither dermatitis nor changes in TEWL levels were observed compared with vehicle-treated mice (Figure 2c,f).

### 2.2. Calcitriol Improves the Symptoms of Atopic Dermatitis in Dfb-Treated NC/Nga Mice

We further investigated whether calcitriol would ameliorate the symptoms of atopic dermatitis in NC/Nga mice treated with Dfb ointment. Increased dermatitis score, TEWL, and epidermal thickness were observed in the Dfb-induced atopic dermatitis model in NC/Nga mice compared with control NC/Nga mice (Figure 3a–c). Dermatitis scores were significantly lower in the 1 nmol calcitriol-treated group than in the vehicle-treated group of Dfb-induced atopic dermatitis model mice (Figure 2a), whereas treatment with 0.1 nmol calcitriol did not significantly decrease dermatitis scores (Appendix A). TEWL levels were significantly decreased in the 1 nmol calcitriol-treated group compared with the vehicle- and 0.1 nmol calcitriol-treated groups after 3 days and 4 days of calcitriol treatment (Figure 3b and Appendix A). Using hematoxylin-eosin staining, acanthosis and increased epidermal thickness were observed in NC/Nga mice with atopic dermatitis, and mice treated with 1 nmol calcitriol but not 0.1 nmol calcitriol showed decreased epidermal thickness (Figure 3c,d and Appendix A).

To evaluate the tight junction barrier function, a permeability assay was performed using a paracellular tracer. A membrane-impermeable paracellular tracer (sulfo-NHS-LC-biotin, molecular weight 556.59 Da) was injected into the dermis, and its diffusion was visualized by the detection of fluorophore-conjugated streptavidin, as described previously [10]. Claudin-1 was used as a marker of tight junctions because its presence is a prerequisite for the paracellular barrier function of tight junctions [9,10,11]. In control mice treated with/without calcitriol, the biotin tracer was detected only in the dermis, and the robust expression of claudin-1 was observed in the epidermis (Figure 3e). The expression of claudin-1 was decreased in the Dfb-induced atopic dermatitis model in NC/Nga mice, and sulfo-NHS-LC-biotin diffused into the epidermis of these mice, indicating the dysfunction of tight junction barriers (Figure 3e). The topical application of calcitriol to atopic dermatitis mice enhanced claudin-1 expression and prevented biotin diffusion into the upper layers of the epidermis (Figure 3e).

### 2.3. Effects of Calcitriol on the Expression of Skin Barrier-Related Proteins

The expression of skin barrier-related proteins is decreased in NC/Nga mice with atopic dermatitis [31,32]. Because calcitriol alleviated skin barrier dysfunction, the epidermal expression of skin barrier-related proteins was examined by immunohistochemistry. The expression of filaggrin, involucrin, and loricrin was decreased in the epidermis of NC/Nga mice with atopic dermatitis. Importantly, calcitriol application increased the downregulated expression of these proteins in mice with atopic dermatitis (Figure 4a). At the mRNA expression level, we observed that only *involucrin* was markedly downregulated in NC/Nga mice with atopic dermatitis compared to the control group, and its expression was significantly restored by calcitriol application (Figure 3c). In NC/Nga mice with atopic dermatitis, calcitriol also substantially enhanced the mRNA expression of both *filaggrin* and *loricrin* (Figure 4b,d). In normal control mice, calcitriol significantly elevated *loricrin* expression and tended to increase *filaggrin* expression (Figure 4b,d).

### 2.4. Effects of Calcitriol on the Expression of Tight Junction-Related Proteins

The decreased expression of tight junction proteins such as occludin and ZO-1 has been reported in NC/Nga mice with atopic dermatitis [31,32,33]. Therefore, we examined the effect of calcitriol on the expression of tight junction-related proteins, including claudin-1, occludin, and ZO-1. We observed a decreased expression of claudin-1, occludin, and ZO-1 in the epidermis of NC/Nga mice with atopic dermatitis compared with control mice (Figure 5a). Interestingly, the administration of calcitriol restored the expression of these tight junction-related proteins (Figure 5a). At the mRNA level, calcitriol enhanced the expression of *claudin-1*, *occludin*, and tight junction protein 1 (*Tjp1*, gene encoding ZO-1) in NC/Nga mice with atopic dermatitis (Figure 5b–d). Among these tight junction components, only the mRNA expression of *occludin* was decreased in NC/Nga mice with atopic dermatitis compared to normal mice (Figure 5c).

### 2.5. Effects of Calcitriol on the Expression of Skin-Derived Antimicrobial Peptides in NC/Nga Mice

Calcitriol is a potent inducer of various skin-derived antimicrobial peptides [16,34]; however, its effect on the expression of these molecules in the skin of NC/Nga mice is unclear. We therefore investigated the effects of calcitriol on the expression of antimicrobial peptides in the skin of NC/Nga mice. Two major groups of antimicrobial peptides, namely, β-defensins and cathelicidins, have been well characterized in human skin, and their mouse homologs have been reported [35,36]. The mRNA expression of β-defensin-1 (*Defb1*) and *Defb2* was upregulated by calcitriol treatment in NC/Nga mice with atopic dermatitis (Figure 6a,b). In addition, the mRNA expression of *Defb3* and *Defb14* was increased in untreated NC/Nga mice with atopic dermatitis, and the topical application of calcitriol did not further elevate this expression (Figure 6c,d). In normal control mice, calcitriol significantly induced *Defb14* expression (Figure 6d) and tended to elevate the expression of *Defb1* (Figure 6a). The expression of cathelicidin antimicrobial peptide (*Camp*) tended to increase in NC/Nga with atopic dermatitis, and calcitriol had no significant effect on it (Figure 6e).

### 2.6. Effects of Calcitriol on Cytokine Expression in the Skin of NC/Nga Mice

To clarify the effect of calcitriol on the expression of atopic dermatitis-related inflammatory cytokines, the expression of inflammatory cytokines in the skin of NC/Nga mice was examined. We observed an increased mRNA expression of type 2 inflammatory mediators, including interleukin-4 (*Il4*), *Il13*, and *Il33*, in the skin of NC/Nga mice with atopic dermatitis compared with control mice (Figure 7a–c). In NC/Nga mice with atopic dermatitis, the expression of *Il4* did not change between the vehicle- and calcitriol-treated groups (Figure 7a). Calcitriol administration attenuated the expression of *Il13* and *Il33* in the skin of NC/Nga mice with atopic dermatitis compared to vehicle-treated mice (Figure 7b,c). Calcitriol significantly upregulated thymic stromal lymphopoietin (*Tslp*) in normal control mice but did not affect *Tslp* mRNA expression in the atopic skin of NC/Nga mice (Figure 7d).

## 3. Discussion

This study describes that calcitriol, an active form of vitamin D3, improves skin barrier dysfunction in atopic dermatitis by inducing skin barrier-related protein expression, alleviating dermatitis and regulating skin-derived antimicrobial peptides. These results suggest that calcitriol might be effective for the treatment of atopic dermatitis.

First, we sought to identify a suitable duration and dose of calcitriol application using both NC/Nga mice and C57BL/6J mice because a mouse model of acute atopic dermatitis is often induced in C57BL/6 mice by the long-term application of calcipotriol, a synthetic derivative of calcitriol, also called MC903 [25,26,27,28,29,30]. In the MC903-induced dermatitis model, the ears of C57BL/6 mice were topically administered with 1 nmol MC903 for 12 days [26], 2 nmol MC903 for 10 days [27], 2 nmol MC903 for 9 days [28], and 2 nmol MC903 for 14 days [29]. Reportedly, the topical application of 1 nmol MC903 (Days 0, 2, and 5) three times did not increase TEWL levels in C57BL/6 mice on Day 7 [30]. In this study, we observed that the daily topical application of 1 nmol calcitriol for 4 days did not affect TEWL in C57BL/6J mice. This might be explained by the fact that the pathogenic mechanism of dermatitis varies among mouse strains [25]. Interestingly, this study demonstrated that a daily topical application of 1 nmol calcitriol for 4 days improved dermatitis in an atopic dermatitis model of NC/Nga mice. Although the dose of calcipotriol for the treatment of psoriasis is 50 μg/g, a 10-day application of low-dose calcitriol (0.6 μg/g) was reported to improve atopic eyelid dermatitis [23]. These observations suggest that the short-term or low-dose application of vitamin D3 might be beneficial for the treatment of atopic dermatitis.

Calcitriol largely decreased TEWL and improved dermatitis scores in NC/Nga mice with atopic dermatitis, suggesting that calcitriol not only alleviates dermatitis but also improves skin barrier function. Immunohistochemical analysis showed that filaggrin, involucrin, loricrin, claudin-1, occludin, and ZO-1 were decreased in NC/Nga mice with atopic dermatitis, which is consistent with previous reports [31,32]. Importantly, the application of calcitriol to the back skin of NC/Nga mice with atopic dermatitis recovered the downregulation of skin barrier-related proteins. At the mRNA level, *involucrin* and *occludin* were downregulated, while the expression of *filaggrin*, *loricrin, claudin-1*, and *Tjp1* was not changed in NC/Nga mice with atopic dermatitis. This observation supports evidence from a previous study that the filaggrin and loricrin downregulation that characterizes atopic dermatitis in humans is not present in NC/Nga mice with atopic dermatitis [37]. Interestingly, calcitriol application upregulated filaggrin, loricrin, claudin-1, and Tjp1 in NC/Nga mice with atopic dermatitis. In patients with atopic dermatitis, it has been reported that the mRNA expression of *filaggrin*, *loricrin*, *occludin*, and *claudin-1* was decreased, while the expression of *involucrin* and *TJP1* was not affected compared with control subjects [4]. Because skin barrier function involves many molecules, including lipids, natural moisturizing factors, and keratins, variation in the expression of skin barrier-related proteins may be caused by differences in among mouse strains or differences in pathophysiology of human skin [25,37,38]. In fact, the molecular mechanisms of immune polarization and skin barrier dysfunction in atopic dermatitis vary according to ethnic origins, between intrinsic and extrinsic atopic dermatitis, as well as between pediatric and adult atopic dermatitis [1,38]. Therefore, further studies are needed to clarify which types of atopic dermatitis can be treated with vitamin D. In addition, atopic dermatitis in Asian people is highly related to the induction of inflammatory cytokines, such as IL-17A, IL-19, and IL-22 in lesional and/or nonlesional skin compared with European patients with atopic dermatitis, suggesting a unique immune phenotype, combining characteristics of atopic dermatitis and psoriasis [39]. Given the therapeutic benefits of psoriasis treatment, vitamin D may also be beneficial for patients with atopic dermatitis in Asian subpopulations.

To defend against invading pathogens, the skin produces hundreds of antimicrobial peptides. Reportedly, the oral administration of vitamin D increases the expression of the antimicrobial peptide cathelicidin in the lesional skin of patients with atopic dermatitis [40]. Although vitamin D3 has been reported to induce the expression of antimicrobial peptides in human skin [34,41], its effect on mouse antimicrobial peptides has not yet been extensively studied. Hence, we investigated the role of calcitriol on skin-derived antimicrobial peptides in NC/Nga mice. Calcitriol increased the expression of *Defb14* and tended to elevate that of *Defb1* (*p* = 0.0695) in NC/Nga mice without atopic dermatitis, but it failed to elevate the expression of *Defb2*, *Defb3*, and *Camp*. *Defb2* expression has not been previously reported due to the lack of its orthologous gene in humans, while *Defb3* and *Camp* genes have been shown to lack a vitamin D receptor response element (VDRE) in their promoters [42,43]. In the skin of atopic dermatitis, the expression of human β-defensin (hBD)-1, -2, and -3 is elevated compared to the skin of healthy individuals [44,45]. Here, the expression of *Defb3* and *Defb14* (orthologous of hBD-2 and -3, respectively) significantly increased in the skin of untreated NC/Nga mice with atopic dermatitis compared with control mice, while the expression of *Defb1*, a homolog of hBD-1, tended to increase, although not significantly (*p* = 0.1247). The expression of cathelicidin in the skin of patients with atopic dermatitis was reported to increase or not change compared with healthy subjects [44,46]. In our study, the tendency of increasing *Camp* expression was observed in NC/Nga mice with atopic dermatitis (*p* = 0.1070). These findings suggest that the induction of dermatitis in NC/Nga mice affects the expression of antimicrobial peptides, while the effects of calcitriol are difficult to compare between mouse models and humans.

The induction of antimicrobial peptides in patients with atopic dermatitis is both beneficial and harmful. Atopic dermatitis is characterized by excessive inflammation, a dysfunctional skin barrier, and frequent colonization by *Staphylococcus aureus* [1]. Because skin-derived antimicrobial peptides are known to suppress inflammation, improve skin barrier functions, and kill *Staphylococcus aureus* [47,48,49,50], antimicrobial peptides may be therapeutic targets for the treatment of atopic dermatitis. However, antimicrobial peptides have been reported to induce the IL-4, IL-13, and IL-31 inflammatory pathways [19,51,52,53], which are hallmark features of atopic dermatitis, indicating that these peptides might function as a double-edged sword in the pathogenesis of atopic dermatitis. Therefore, the beneficial role of calcitriol in atopic dermatitis treatment through the induction of antimicrobial peptides needs further clarification.

The pathogenesis of atopic dermatitis involves various inflammatory cytokines, particularly IL-4 and IL-13, acting on both immune cells and nonimmune cells and enhancing type 2 inflammation [3]. Skin barrier defects also lead to the production of canonical type 2 cytokines from immune cells and to the release of alarmins, such as IL-33 and TSLP, from keratinocytes [3]. The expression of *Il4*, *Il13*, and *Il33* was increased in the skin of NC/Nga mice with atopic dermatitis, while calcitriol application decreased *Il13* and *Il33* mRNA expression. IL-33 belongs to the IL-1 family, which induces type 2 innate lymphoid cells (ILC2s) to release type 2 cytokines, particularly IL-13 and IL-5 [54]. These results suggest that restoring epidermal barrier function by calcitriol may induce a decrease in the expression of IL-13 and IL-33. Furthermore, TSLP promotes type 2 immune responses that have been reported to be the main mediator of inflammation in MC903-induced dermatitis model mice [25]. In addition, the skin of MC903-induced dermatitis model mice showed increased expression of *Il33* [26]. Calcitriol application induced an increased level of *Tslp* expression but did not affect the level of *Il33* expression in control NC/Nga mice. These results may indicate that the immune response of the skin depends on the mouse strain.

Calcitriol has been suggested to alleviate psoriasis by inhibiting keratinocyte proliferation, promoting cell differentiation, improving skin barrier function, and modulating immune responses [55]. Here, we provide evidence that calcitriol application might improve symptoms in atopic dermatitis by improving skin barrier function and regulating antimicrobial peptides. However, because vitamin D3 induces hypercalcemia and dermatitis [16], short-term application or proactive treatment with vitamin D3 might be an appropriate strategy for the treatment of atopic dermatitis. These results suggest that calcitriol might be a therapeutic agent for the treatment of atopic dermatitis in addition to psoriasis.

## 4. Materials and Methods

### 4.1. Reagents

Calcitriol, ethanol, and 4′,6-diamidino-2-phenylindole dihydrochloride n-hydrate (DAPI) were obtained from Fujifilm Wako Pure Chemical Corporation (Osaka, Japan). Dfb ointment (Biostir-AD) was purchased from Biostir Inc. (Osaka, Japan).

Primary antibodies against filaggrin (1:1000 dilution), involucrin (1:1000 dilution), and loricrin (1:1000 dilution) were purchased from BioLegend (San Diego, CA, USA). Anti-claudin-1 antibody (1:20,000 dilution) was obtained from Spring Bioscience (Pleasanton, CA, USA), anti-occludin antibody (1:1000 dilution) was obtained from Cell Signaling Technology (Beverly, MA, USA), and anti-ZO-1 antibody (1:4000 dilution) was purchased from Proteintech (Rosemont, IL, USA). The secondary antibodies conjugated to Alexa Fluor dye were purchased from Thermo Fisher Scientific (Waltham, MA, USA) and diluted 1:1000.

### 4.2. Animals

Male NC/Nga mice and C57BL/6J mice aged 4–7 weeks were purchased from Sankyo Labo Service Corporation Inc. (Tokyo, Japan) and kept in an animal room under SPF conditions with controlled temperature (23 ± 2 °C), humidity (50 ± 10%), and light (lights on from 8:00 to 20:00). Food and tap water were provided ad libitum. The average body weight was 22.5 ± 1.6 g for NC/Nga mice and 22.5 ± 1.1 g for C57BL/6J mice. All animal procedures were approved by the Institutional Animal Care and Use Committee at Juntendo University Graduate School of Medicine and conformed to the Guidelines for the Use of Laboratory Animals of the National Institutes of Health.

### 4.3. Induction of Dermatitis and Treatment of Mice

Dermatitis was induced in NC/Nga mice as described [56,57]. Briefly, the rostral part of each mouse aged 8 weeks was shaved with an electric shaver, and the residual hair in this area was removed using a hair removal cream. Then, 100 mg of Dfb ointment was applied topically to the shaved area of each mouse (antigen sensitization). Twice weekly for three weeks, the growing hair was shaved, followed by application of 4% sodium dodecyl sulfate (SDS; 150 μL/site) on the treated area. Dfb ointment (100 mg/site) was topically applied two hours after barrier disruption. After three weeks of Dfb application on Day 21, mice with dermatitis scores above 6 were selected.

Beginning when the mice were 11 weeks old, 1 nmol calcitriol in 150 μL ethanol or a vehicle control was applied to the dorsal skin of each mouse once daily for 4 days.

### 4.4. Evaluation of Skin Lesions and Condition

Before each treatment, dorsal lesions were evaluated for four symptoms: (1) erythema/hemorrhage, (2) scarring/dryness, (3) edema, and (4) excoriation/erosion. Each symptom was graded from 0 to 3 (none, 0; mild, 1; moderate, 2; and severe, 3). The sum of the individual scores was taken as the dermatitis score, ranging from 0 to 12. TEWL in dorsal skin was measured using a VapoMeter (Delfin Technologies Ltd., Kuopio, Finland).

### 4.5. Histological and Immunohistochemical Analysis

Dorsal skin was excised and fixed with 20% formalin solution (Fujifilm Wako Pure Chemical Corporation). The skin samples were embedded in paraffin by a conventional method and stained with hematoxylin and eosin. Images were acquired using an Olympus virtual slide system VS110 (Olympus, Tokyo, Japan). The thickness of the epidermis was expressed as the mean values from five fields per mouse.

For immunohistochemistry, cryosections of mouse dorsal skin were fixed with 4% paraformaldehyde (Fujifilm Wako Pure Chemical Corporation) for 10 min, washed with 0.1% Triton X-100 in phosphate-buffered saline for 10 min, and incubated with Blocking One (Nacalai Tesque, Kyoto, Japan) for 30 min. Following overnight incubation at 4 °C with primary antibodies, the sections were incubated with Alexa Fluor-conjugated secondary antibodies at room temperature for 60 min in the dark and mounted in PLUS Antifade Mounting Medium (Vector Laboratories, Newark, CA, USA). Antibodies were diluted with Signal Enhancer HIKARI (Nacalai Tesque). Images were acquired using an LSM 700 confocal laser microscope (Carl Zeiss Co., Ltd., Oberkochen, Germany).

### 4.6. Tight Junction Permeability Assay

The assay was performed according to a previously described method [10]. A volume of 50 μL of 10 mg/mL EZ-Link™ Sulfo-NHS-LC-Biotin (Thermo Fisher Scientific) in phosphate-buffered saline containing 1 mM CaCl_2_ was injected into the dermis of the back skin. After 30 min, skin samples were collected and frozen in liquid nitrogen. Frozen sections were fixed and soaked in blocking solution as described for immunohistochemistry. Alexa Fluor 594-conjugated streptavidin (1:10,000 dilution; Thermo Fisher Scientific) was incubated with secondary antibodies.

### 4.7. Total RNA Preparation and Quantitative Real-Time PCR Analysis

Total RNA from mouse skin was extracted using the RNeasy Plus Universal Mini kit (Qiagen, Tokyo, Japan). Reverse transcription reactions were performed with ReverTra Ace qPCR RT Master Mix with gDNA Remover (TOYOBO, Osaka, Japan) according to the manufacturer’s protocol. Quantitative real-time PCR was performed on a StepOnePlus Real-Time PCR system (Applied Biosystems, Foster City, CA, USA) using a QuantiNova SYBR Green PCR Kit (Qiagen). Each sample was analyzed in duplicate; the amounts of mRNA were normalized to those of β-actin (*Actb*) and, finally, presented as ratios to those of the vehicle group. The primers used in this study are provided in Table 1.

### 4.8. Statistical Analysis

The data were analyzed using GraphPad Prism 8 (GraphPad Software Inc., San Diego, CA, USA). The differences between the means were analyzed by one-way or two-way ANOVA with Tukey’s or Sidak’s multiple comparison test. In all analyses, *p* < 0.05 was considered statistically significant.

## Figures and Tables

**Figure 1 ijms-24-09347-f001:**
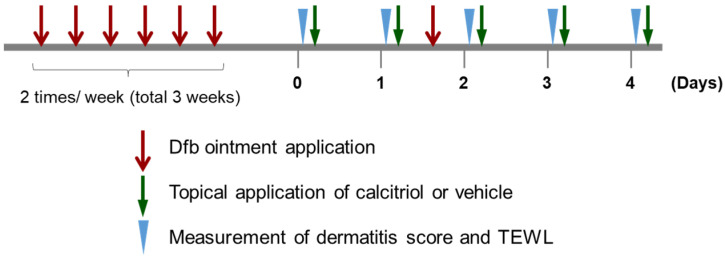
Experimental protocol for induction of the atopic dermatitis model and treatment with calcitriol. Dermatitis was induced in NC/Nga mice aged 8 weeks following a topical application of Dfb ointment twice weekly for 3 weeks. Calcitriol and vehicle were applied once daily for 4 days.

**Figure 2 ijms-24-09347-f002:**
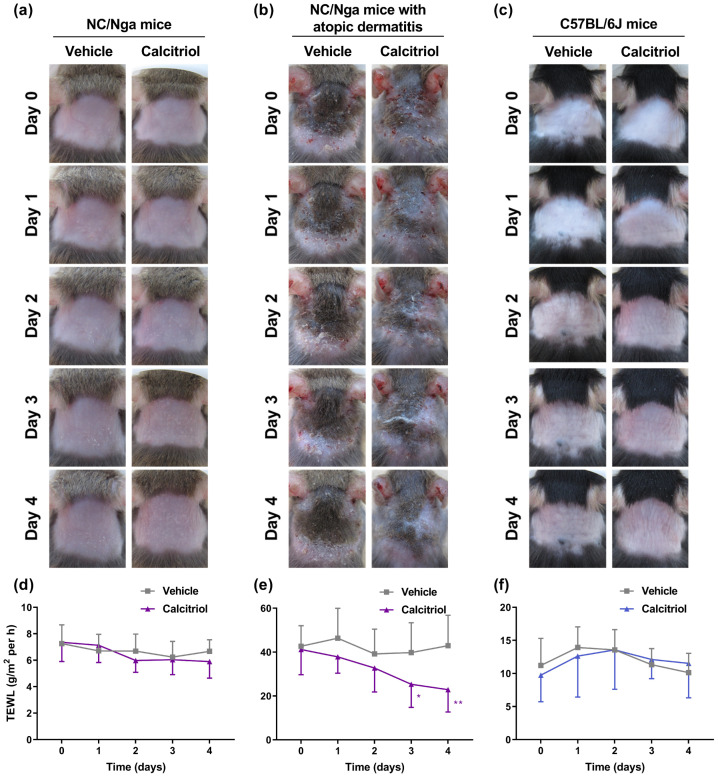
The effects of a daily topical application of calcitriol for 4 days on the condition of the skin in NC/Nga mice without atopic dermatitis, NC/Nga mice with atopic dermatitis, and C57BL/6J mice. (**a**–**c**) Representative images of the back skin treated with vehicle (left panels) and 1 nmol calcitriol (right panels) in NC/Nga mice (**a**), NC/Nga mice with atopic dermatitis (**b**), and C57BL/6J mice (**c**). (**d**–**f**) TEWL data represent the means ± standard deviations (SDs) for the back skin of NC/Nga mice without atopic dermatitis ((**d**), n = 10), NC/Nga mice with atopic dermatitis ((**e**), n = 9), and C57BL/6J mice ((**f**), n = 5). * *p* < 0.05 and ** *p* < 0.01 compared with the vehicle group at each time point using two-way ANOVA with Sidak’s multiple comparison test.

**Figure 3 ijms-24-09347-f003:**
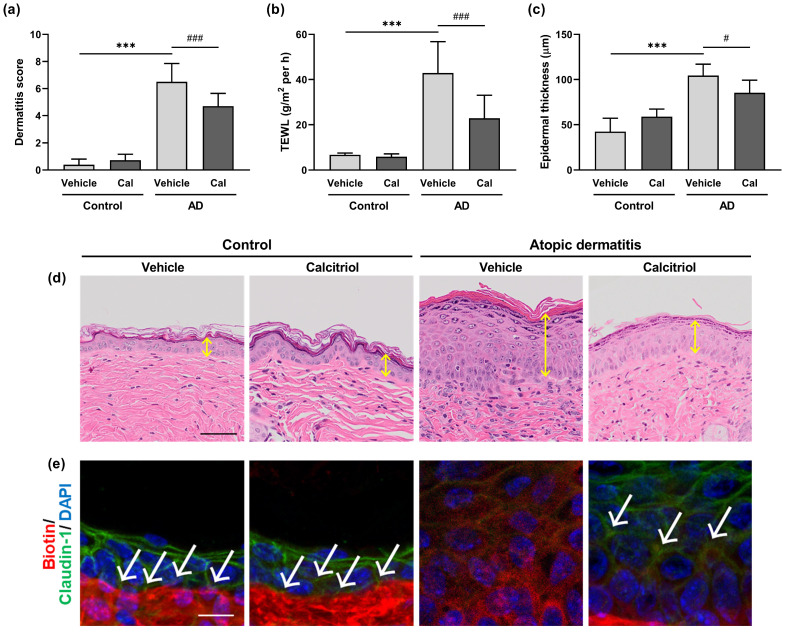
Calcitriol improves the symptoms of atopic dermatitis in NC/Nga mice. (**a**–**c**) Dermatitis score (**a**), TEWL (**b**), and epidermal thickness on Day 4 in vehicle-treated NC/Nga mice (control–vehicle), calcitriol-treated NC/Nga mice (control–Cal), vehicle-treated atopic NC/Nga mice (AD–vehicle) and calcitriol-treated atopic NC/Nga mice (AD–Cal). The data represent the mean ± SD in each group (n = 9). *** *p* < 0.001 compared between the control–vehicle and AD–vehicle groups; # *p* < 0.05 and ### *p* < 0.001 compared between the AD–vehicle and AD–Cal groups by one-way ANOVA with Tukey’s multiple comparison test. (**d**) Representative hematoxylin-eosin staining images of the back skin on Day 4. The yellow double arrows indicate the epidermis. Scale bar = 50 μm. (**e**) Calcitriol enhances tight junction barrier function in atopic dermatitis mice. Representative immunofluorescence images of the tight junction permeability assay on Day 4. Sulfo-NHS-LC-biotin (Biotin; red) and claudin-1 (green) were made visible with Alexa Fluor dyes, and DAPI (blue) served as a nuclear counterstain. Tracer exclusion is indicated by the white arrowheads. Scale bar = 10 μm.

**Figure 4 ijms-24-09347-f004:**
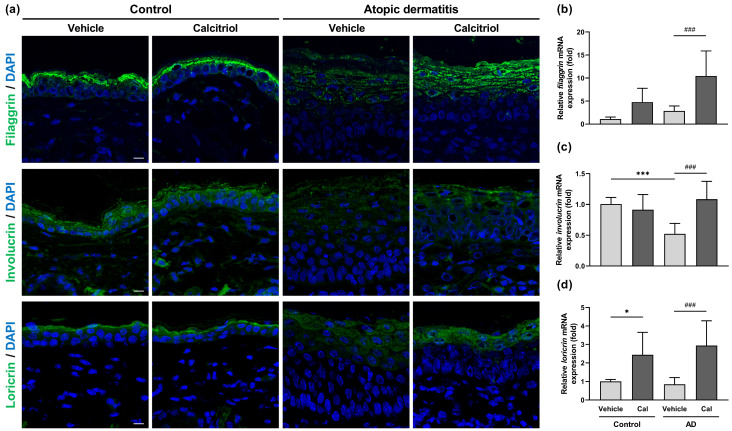
Calcitriol upregulates the expression of skin barrier-related proteins, filaggrin, involucrin, and loricrin in NC/Nga mice with atopic dermatitis. (**a**) Immunocytochemistry showing the epidermal expression of filaggrin, involucrin, and loricrin on Day 4 in the dorsal skin. DAPI served as a nuclear counterstain. Scale bar = 10 μm. (**b**–**d**) Expression of *filaggrin* (**b**), *involucrin* (**c**), and *loricrin* (**d**) mRNAs by quantitative real-time PCR in the back skin of vehicle-treated NC/Nga mice (control–vehicle), calcitriol-treated NC/Nga mice (control–Cal), vehicle-treated atopic dermatitis mice (AD–vehicle), and calcitriol-treated mice with atopic dermatitis (AD–Cal). The data represent the mean ± SD in each group (n = 6–9). * *p* < 0.05 and *** *p* < 0.001 compared with the control–vehicle group; ### *p* < 0.001 compared between the AD–vehicle and AD–Cal groups by one-way ANOVA with Tukey’s multiple comparison test.

**Figure 5 ijms-24-09347-f005:**
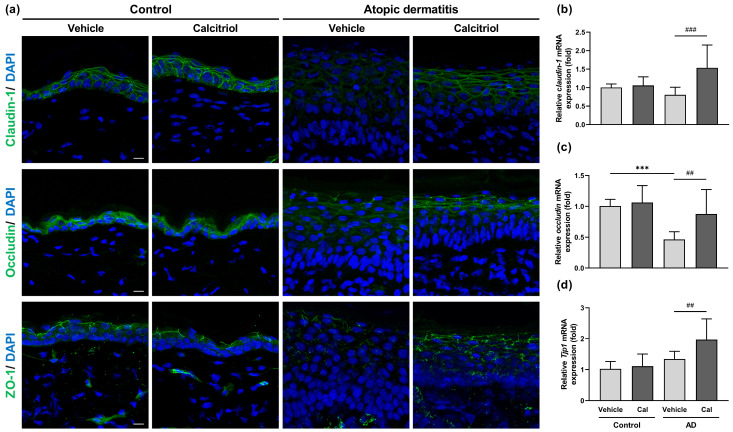
Calcitriol upregulates the expression of tight junction-related proteins, claudin-1, occludin, and ZO-1 in NC/Nga mice with atopic dermatitis. (**a**) Immunocytochemistry showing the epidermal expression of claudin-1, occludin, and ZO-1 on Day 4 in the dorsal skin. DAPI served as a nuclear counterstain. Scale bar = 10 μm. (**b**–**d**) Expression of *claudin-1* (**b**), *occludin* (**c**), and *Tjp1* (**d**) mRNAs by quantitative real-time PCR in the back skin of vehicle-treated NC/Nga mice (control–vehicle), calcitriol-treated NC/Nga mice without atopic dermatitis (control–Cal), vehicle-treated atopic dermatitis mice (AD–vehicle), and calcitriol-treated NC/Nga mice with atopic dermatitis (AD–Cal). The data represent the mean ± SD in each group (n = 6–9). *** *p* < 0.001 compared between the control–vehicle and AD–vehicle groups; ## *p* < 0.01 and ### *p* < 0.001 compared between the AD–vehicle and AD–Cal groups by one-way ANOVA with Tukey’s multiple comparison test.

**Figure 6 ijms-24-09347-f006:**
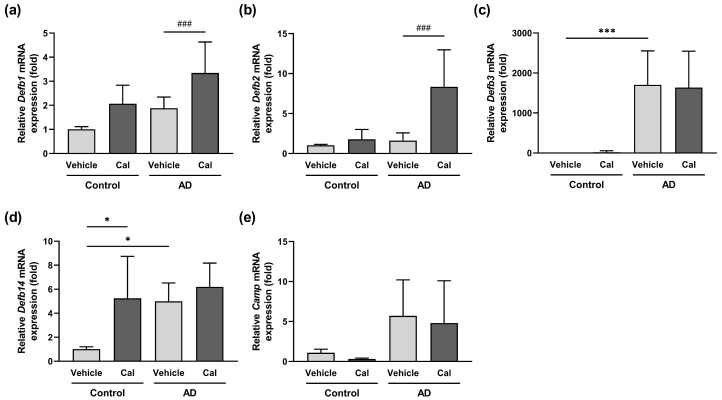
Effects of calcitriol on the expression of skin-derived antimicrobial peptides in NC/Nga mice. The mRNA expression was analyzed by quantitative real-time PCR on Day 4 in the dorsal skin of vehicle-treated NC/Nga mice (control–vehicle), calcitriol-treated NC/Nga mice without atopic dermatitis (control–Cal), vehicle-treated mice with atopic dermatitis (AD–vehicle), and calcitriol-treated NC/Nga mice with atopic dermatitis (AD–Cal). The mRNA levels of the antimicrobial peptides *Defb1* (**a**), *Defb2* (**b**), *Defb3* (**c**), *Defb14* (**d**), and *Camp* (**e**) were quantified. The data represent the mean ± SD in each group (n = 5–9). * *p* < 0.05 and *** *p* < 0.001 compared with the control–vehicle group; ### *p* < 0.001 compared between the AD–vehicle and AD–Cal groups by one-way ANOVA with Tukey’s multiple comparison test.

**Figure 7 ijms-24-09347-f007:**
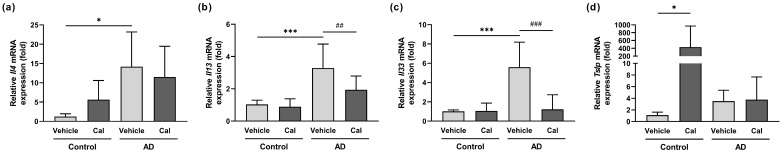
Effects of calcitriol on the expression of type 2 inflammatory mediators in NC/Nga mice. The mRNA expression was analyzed by quantitative real-time PCR on Day 4 in the dorsal skin of vehicle-treated NC/Nga mice (control–vehicle), calcitriol-treated NC/Nga mice without atopic dermatitis (control–Cal), vehicle-treated mice with atopic dermatitis (AD–vehicle), and calcitriol-treated NC/Nga mice with atopic dermatitis (AD–Cal). The mRNA levels of *Il4* (**a**), *Il13* (**b**), *Il33* (**c**), and *Tslp* (**d**) were quantified. The data represent the mean ± SD in each group (n = 4–8). * *p* < 0.05 and *** *p* < 0.001 compared with the control–vehicle group; ## *p* < 0.01 and ### *p* < 0.001 compared between the AD–vehicle and AD–Cal groups by one-way ANOVA with Tukey’s multiple comparison test.

**Table 1 ijms-24-09347-t001:** Primers used for quantitative real-time PCR.

Gene	Primer	Sequence (5′–3′)
*Filaggrin*	Forward	CATCCGTAAAGACCTCTATGCCAAC
	Reverse	ATGGAGCCACCGATCCACA
*Involucrin*	Forward	AACCACACCAGTGTCCCAGAG
	Reverse	TGGGTGAGTAGGCCAGCTGA
*Loricrin*	Forward	CTCACTCATCTTCCCTGGTGCTT
	Reverse	CCTCCTCCACCAGAGGTCTTTC
*Claudin-1*	Forward	ACCGGGCAGATACAGTGCAA
	Reverse	TGCCAATGGTGGACACAAAGA
*Occludin*	Forward	GGCAAGCGATCATACCCAGAG
	Reverse	AGGCTGCCTGAAGTCATCCAC
*Tjp1*	Forward	GTTGGTACGGTGCCCTGAAAGA
	Reverse	GCTGACAGGTAGGACAGACGAT
*Defb1*	Forward	AGCCTCATCTGTCAGCCCAACTA
	Reverse	TCCAAGACTTGTGAGAATGCCAAC
*Defb2*	Forward	TTTCTACCAGCCATGAGGACTCT
	Reverse	CAGTGGTCAAGTTCTGCTTCGT
*Defb3*	Forward	TCAGTCATGAGGATCCATTACCTTC
	Reverse	GCCAATGCACCGATTCCAG
*Defb14*	Forward	ATCTTGTTCTTGGTGCCTGC
	Reverse	CTTCTTTCGGCAGCATTTTC
*Camp*	Forward	TGCTCCGAGCTGTGGATGAC
	Reverse	CCTTCACTCGGAACCTCACAGAC
*Il4*	Forward	ACGGAGATGGATGTGCCAAAC
	Reverse	AGCACCTTGGAAGCCCTACAGA
*Il13*	Forward	CGGCAGCATGGTATGGAGTG
	Reverse	ATTGCAATTGGAGATGTTGGTCAG
*Il33*	Forward	GAGACTCCGTTCTGGCCTCA
	Reverse	AATGTGTCAACAGACGCAGCAA
*Tslp*	Forward	CGAGCAAATCGAGGACTGTGAG
	Reverse	GCAGTGGTCATTGAGGGCTTC
*Actb*	Forward	CATCCGTAAAGACCTCTATGCCAAC
	Reverse	ATGGAGCCACCGATCCACA

## Data Availability

The data presented in this study are available upon request from the corresponding author.

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
