# Peer review of "Calcitriol, an Active Form of Vitamin D3, Mitigates Skin Barrier Dysfunction in Atopic Dermatitis NC/Nga Mice"

_ijms, 2023, doi:10.3390/ijms24119347_

Round 1

Reviewer 1 Report

This is well designed manuscript which for the first time shown that topical calcitriol application to mouse with atopic dermatitis bringing beneficial effects. While other studies showed that topical calcitriol application exacerbated AD both in mouse models and human case reports due to induction of TSLP or IL-4, authors found something different. Well, these opposite effects might be due to different mouse strains used for AD induction, different AD models, calcitriol concentration etc.

Minor:

Please cited and discussed accordingly one case report in the manuscript:

JAAD Case Rep . 2018 Nov 27;5(1):5-6. doi:10.1016/j.jdcr.2018.09.012.

Major:

I would recommend checking the expression of mentioned cytokines i.e., TSLP and IL-4 at protein or mRNA levels in skin biopsies of non- and treated mouse.

Reviewer 2 Report

In the manuscript, Umehara et al. attempted to investigate the effect of calcitriol on skin barrier dysfunction in atopic dermatitis NC/Nga mice. The topic is interesting, and experimental design is straightforward. In order to improve the quality of the manuscript, several issues need to be improved, as follows:

1.     Line 15-19 Please shorten down the background of the study in abstract section.

2.     The main technique applied in the study should be indicated in Keywords.

3.     Line 84-89 Please do not report results in the introduction section.

4.     In the result section, figures should be appeared following the referring sentences subsequently. Please modify through the entire manuscript.

5.     In Figure 2 (d) and (f), please modify the scale of y axis, so as to improve the readability.

6.     Line 210-215 Such sentences should be moved to discussion section.

7.     Why did the authors just report exact p value for Figure 6 (a) and (e), instead for all figures?

8.     Line 335 The average weight of animals should be reported.

9.     Line 352 150 μl should be 150 μL.

10.   Line 397 P should be italic.

Minor editing of English language required.

Round 2

Reviewer 1 Report

Thank you for considering all my suggestions in the revised manuscript. I have no further comments.